# NONNEGATIVE MATRIX FACTORIZATION THROUGH CANONICAL EDGES

## ABSTRACT

In this paper we present a novel approach to *nonnegative matrix factorization* (NMF) by introducing the concept of *nonnegative canonical edges* (NCEs). These NCEs are intersections of the principal subspace containing the data to be factored with canonical faces of the nonnegative orthant. Through this lens, our approach yields a closed-form solution to the special NMF case where (at least one of) the factors are required to be orthogonal. In the general NMF case, NCEs provide a deterministic optimal solution whenever the data resides within or in proximity to the cone formed by the NCEs. Furthermore, NCEs provide an improved initialization for classical NMF methods in general. Despite these advancements, numerous fundamental questions regarding NCEs in the context of NMF remain unexplored, offering exciting avenues for future research.

## 1 INTRODUCTION

Decomposing a data matrix $\mathbf{X}$ into the product of two nonnegative factors $\mathbf{W}$ and $\mathbf{H}$ has become a fundamental problem in data science. This problem, known as *nonnegative matrix factorization* (NMF), has an endless list of applications in computer vision, spectral data analysis, bioinformatics, chemometrics, document classification, clustering, feature selection, and many more (Li & ZHANG, 2008; Wang & Zhang, 2012; Huang et al., 2012; Del Buono & Pio, 2015; vCopar et al., 2019; Li & Ding, 2018; Ang & Gillis, 2019; Gan et al., 2021; Lee & Roy, 2021; Baur et al., 2022; Chen et al., 2022). The main appeal of NMF over other factorizations like *principal component analysis* (PCA) (Pearson, 1901; Hotelling, 1933; Jolliffe, 2002) is its ability to generate meaningful decompositions where the effect of each component can be readily interpreted. However, even though a myriad of NMF approaches have been proposed over the last decades, current methods only guarantee local convergence (Asteris et al., 2015; Fathi Hafshejani & Moaberfard, 2022).

**Contributions.** This paper introduces a new perspective on NMF that induces a deterministic algorithm guaranteed to successfully factorize data $\mathbf{X}$ satisfying what we call the *canonical inclusion property* (CIP). The main idea is to intersect the subspace $\mathcal{W}$ containing the columns of $\mathbf{X}$ with canonical faces of the nonnegative orthant to obtain what we call the *nonnegative canonical edges* (NCEs) of $\mathcal{W}$, which form the basis of our factorization. The key intuition behind NCEs is to find $\mathrm{r} := \dim(\mathcal{W})$ basis vectors of $\mathcal{W}$ that lie at the boundary of the nonnegative orthant, so that their cone covers as much as possible of the nonnegative portion of $\mathcal{W}$, which we call $\mathcal{W}_+$ (see Figure 1 to build some intuition). We say that data $\mathbf{X}$ satisfies the CIP if and only if it lies in one of the cones defined by $\mathrm{r}$ NCEs (see Figure 2). Whether this is the case depends both on the orientation of the subspace $\mathcal{W}$ and on the distribution of points (columns of $\mathbf{X}$) over such subspace. In general, each NCE cone is a subset of $\mathcal{W}_+$. The wider the NCE cones, the more disperse the columns of $\mathbf{X}$ are allowed to be over $\mathcal{W}_+$ and still satisfy the CIP. The orientation of $\mathcal{W}$ determines the number and location of its NCEs, and in turn, the volume of its NCE cones. Several fundamental questions immediately arise:

(**Q1**) How many NCEs does a given subspace $\mathcal{W}$ have?

(**Q2**) What is the volume of the largest NCE cone relative to $\mathcal{W}_+$?

(**Q3**) Is there an efficient and provable algorithm to identify NCEs and the largest NCE cone?

We give exact answers to these questions in the special case where $\mathbf{X}$ admits a factorization with at least one orthogonal factor (we assume without loss of generality that it is $\mathbf{W}$; otherwise we can

simply transpose $\mathbf{X}$). We show that in this *orthogonal NMF* (ONMF) case, there will be exactly r NCEs, and that their cone is equal to all $\mathcal{W}_+$. As a direct consequence, any $\mathbf{X}$ that admits an ONMF will directly satisfy the CIP, independently of the distribution of its columns over $\mathcal{W}_+$. Using these insights, we obtain a closed-form ONMF solution. In doing so, we provide, to the best of our knowledge, the first ONMF method with global guarantees, thus answering an over 20-year-old open question (Li et al., 2001; Yuan & Oja, 2005; Ding et al., 2006). What is more, our solution only requires computing a projection operator, making it nearly as efficient as an eigenvalue decomposition. We generalize this exact solution to the noisy case, taking advantage of the side information induced by the orthogonality and nonnegative constraints to obtain an optimal solution that it is even more accurate than the unconstrained solution (e.g., obtained by PCA).

As for the general NMF case, we provide partial answers. Namely, we give bounds on the number of NCEs, and empirical evidence that these bounds are loose in the sense that there are typically very few NCEs. This is quite convenient because it narrows down the search space of NCE cones. We also show results suggesting that NCE cones generally cover large portions of $\mathcal{W}_+$, and that the union of all NCE cones is equal to $\mathcal{W}_+$. This suggest that a few additional NCEs would allow factorizing data that does not satisfy the CIP, or data whose nonnegative rank is higher than its rank (Dong et al., 2008). We give simple strategies that use NCEs in the presence of noise, and enjoy optimal accuracy when data satisfies the CIP. On the other hand, we show that if data violates the CIP and NCEs cannot be used directly, they still make good initializations for classical NMF optimization methods, enabling convergence to lower minima.

**Open Questions.** The main remaining challenge is that in general, finding NCEs requires identifying the faces that contain them, which can be a daunting combinatorial problem, especially in adversarial settings. In light of this, two key theoretical results that would advance significantly this NCE approach are: tighter bounds on the number of NCEs, and more importantly, a characterization of the canonical faces containing NCEs as a function of the position of $\mathcal{W}$ relative to the canonical axes. This in turn might enable an efficient, or even direct mechanism to identify NCEs. From a practical standpoint, efficient algorithms or heuristics to find NCEs (analogous, for example, to EVA for extreme vectors (Klingenberg et al., 2009)) could also represent a significant contribution to this methodology.

**Organization of the Paper.** The rest of the paper is organized as follows. Section 2 formally defines NCEs and the CIP. Section 3 discusses their connection to previous work. Section 4 gives evidence that NCE cones cover large portions of $\mathcal{W}_+$. Sections 5 and 6 discusses simple strategies to generalize NCEs to noisy settings, and use them as initialization whenever data does not satisfy the CIP. Le pièce de résistance discussing ONMF is in Section 7.

## 2  MAIN IDEAS AND CONCEPTS

### EXACT NMF

Let $\mathbf{X} \in \mathbb{R}^{m \times n}$ be a rank-r data matrix with nonnegative entries. Suppose there exist matrices $\mathbf{W}^\star \in \mathbb{R}^{m \times r}$ and $\mathbf{H}^\star \in \mathbb{R}^{r \times n}$ with nonnegative entries such that

$$\mathbf{X} = \mathbf{W}^\star \mathbf{H}^\star. \tag{1}$$

Given $\mathbf{X}$, the goal of NMF is to find matrices $\mathbf{W} \in \mathbb{R}^{m \times r}$ and $\mathbf{H} \in \mathbb{R}^{r \times n}$ with nonnegative entries such that $\mathbf{X} = \mathbf{W}\mathbf{H}$. Recall that in general, the *true* pair $(\mathbf{W}^\star, \mathbf{H}^\star)$ is unidentifiable due to the ill-posedness of NMF (Donoho & Stodden, 2003; Dong et al., 2008; Klingenberg et al., 2009).

### NONNEGATIVE CANONICAL EDGES (NCES)

At the heart of our contributions lie what we call *nonnegative canonical edges* (NCEs), which we define as the 1-dimensional intersections of $\mathcal{W} := \mathrm{span}(\mathbf{W}^\star) = \mathrm{span}(\mathbf{X})$ with the canonical faces of the nonnegative orthant. Intuitively, the goal we aim to achieve through NCEs is to find a basis of $\mathcal{W}$ that lies at the boundary of the nonnegative orthant, so that its cone covers as much as possible of $\mathcal{W}_+$, and captures all the data $\mathbf{X}$. To formally define NCEs, let $\Omega$ be a subset of $\{1, \ldots, m\}$ indexing a subset of coordinates. Given $\Omega$, define the *canonical face* $\mathcal{F}_\Omega$ as the span of the canonical vectors $\mathbf{e}_i$ indexed by $\Omega$ (see Figure 1 to build some intuition). Let $\mathbf{I}$, $\mathbf{P}_\mathcal{W}$, and $\mathbf{P}_\Omega$ denote the identity matrix

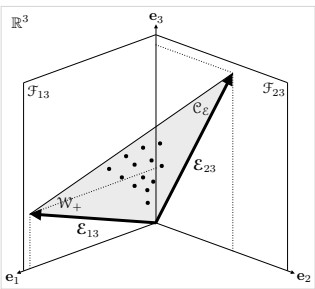

**Figure 1:** $\mathcal{W}_+$ represents the nonnegative subset of the subspace $\mathcal{W}$ containing the columns of $\mathbf{X}$, represented by points. $\mathcal{F}_\Omega$ is the canonical face spanned by the canonical vectors indexed in $\Omega$. In this figure, $\Omega = \{1, 3\}$ produces the face $\mathcal{F}_{13}$ given by the $(x, z)$-plane, and similarly for $\Omega = \{2, 3\}$. The NCE $\mathcal{E}_\Omega$ is the nonnegative vector at the intersection of $\mathcal{W}$ and $\mathcal{F}_\Omega$. $\mathcal{C}_\mathcal{E}$ is the cone generated by the NCEs $\{\mathcal{E}_{13}, \mathcal{E}_{23}\}$. In this $\mathbb{R}^3$ example, $\mathcal{C}_\mathcal{E} = \mathcal{W}_+$. This is also the case when $\mathbf{X}$ admits an orthogonal decomposition (see Section 7). In general, however, $\mathcal{C}_\mathcal{E} \subset \mathcal{W}_+$ (see Figure 2).

and the projection operators onto $\mathcal{W}$ and $\mathcal{F}_\Omega$. We say that

$$\mathcal{E}_\Omega \; := \; \mathcal{W} \cap \mathcal{F}_\Omega \; = \; (\mathbf{I} - \mathbf{P}_\Omega \mathbf{P}_\mathcal{W})^\perp$$

is a *canonical edge of* $\mathcal{W}$ if $\dim(\mathcal{E}_\Omega) = 1$. For subspaces in general position, this will be the case if and only if $|\Omega| = m - r + 1$, because then

$$\dim(\mathcal{W} + \mathcal{F}_\Omega) \; = \; \min(m, \dim(\mathcal{W}) + \dim(\mathcal{F}_\Omega)) \; = \; \min(m, r + |\Omega|) \; = \; m,$$

which implies

$$\begin{aligned}
\dim(\mathcal{E}_\Omega) \; &= \; \dim(\mathcal{W} \cap \mathcal{F}_\Omega) \; = \; \dim(\mathcal{W}) + \dim(\mathcal{F}_\Omega) - \dim(\mathcal{W} + \mathcal{F}_\Omega) \\
&= \; r + |\Omega| - m \; = \; 1.
\end{aligned}$$

In full generality, $r \le \dim(\mathcal{W} + \mathcal{F}_\Omega) \le m$, so $\dim(\mathcal{E}_\Omega) \le |\Omega| \le m - r + \dim(\mathcal{E}_\Omega)$. It follows that $\mathcal{E}_\Omega$ can only be a line if

$$1 \; \le \; |\Omega| \; \le \; m - r + 1. \tag{2}$$

Consequently, we never have to consider sets $\Omega$ outside this range. Whenever $\mathcal{E}_\Omega$ is a line, its spanning vector $\boldsymbol{\mathcal{E}}_\Omega \in \mathbb{R}^m$ can be trivially computed as the 1-norm solution to the linear system

$$(\mathbf{I} - \mathbf{P}_\Omega \mathbf{P}_\mathcal{W}) \boldsymbol{\mathcal{E}}_\Omega \; = \; \mathbf{0}. \tag{3}$$

If $\boldsymbol{\mathcal{E}}_\Omega \ge \mathbf{0}$, we say that $\boldsymbol{\mathcal{E}}_\Omega$ is a *nonnegative canonical edge of* $\mathcal{W}$.

CANONICAL INCLUSION PROPERTY (CIP)

Since $\dim(\mathcal{W}) = r$, $\mathcal{W}$ will *cut* the nonnegative orthant in $r$ directions. Consequently, there will be a collection of *at least* $r$ sets $\Omega$ producing NCEs, and *at least* one subset of $r$ such NCEs will be linearly independent. There may be more than one collection of $r$ linearly independent NCEs because in general there are more than $r$ canonical faces, and $\mathcal{W}$ may intersect with each of them to produce an NCE. We say that $\mathbf{X}$ satisfies the *canonical inclusion property* (CIP) if there exists an $m \times r$ matrix $\mathbf{W}_\mathcal{E}$ formed with $r$ linearly independent NCEs such that the columns of $\mathbf{X}$ are contained in the *NCE cone* $\mathcal{C}_\mathcal{E}$ generated by $\mathbf{W}_\mathcal{E}$ (see Figure 2 for some intuition; notice that $\mathbf{X}$ may be contained in more than one NCE cone). Observe that $\mathbf{X}$ will always be contained in $\mathrm{span}(\mathbf{W}_\mathcal{E}) = \mathcal{W}$, because the $r$ columns of $\mathbf{W}_\mathcal{E}$ lie in $\mathcal{W}$ by definition, and are linearly independent by construction. The question is whether $\mathbf{X}$ lies in the cone $\mathcal{C}_\mathcal{E}$. Given $\mathbf{X}$ and $\mathbf{W}_\mathcal{E}$, verifying whether $\mathbf{X} \in \mathcal{C}_\mathcal{E}$ can be trivially done by checking if the coefficients of $\mathbf{X}$ with respect to $\mathbf{W}_\mathcal{E}$ are nonnegative. These coefficients are given by

$$\mathbf{H}_\mathcal{E} \; := \; (\mathbf{W}_\mathcal{E}^\mathsf{T} \mathbf{W}_\mathcal{E})^{-1} \mathbf{W}_\mathcal{E}^\mathsf{T} \mathbf{X}. \tag{4}$$

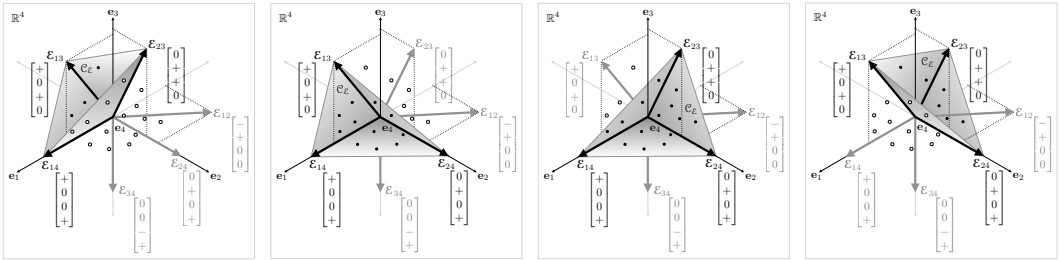

**Figure 2:** Each point in these figures represents a column in $\mathbf{X} \subset \mathbb{R}^4$, viewed orthogonally from the $4^{\text{th}}$ dimension. In this example, the $r = 3$-dimensional subspace $\mathcal{W}$ has $\binom{m}{r-1} = 6$ canonical edges, 4 of which are nonnegative (NCEs): $\{\mathcal{E}_{13}, \mathcal{E}_{14}, \mathcal{E}_{23}, \mathcal{E}_{24}\}$. Therefore, there are 4 NCE cones $\mathcal{C}_{\mathcal{E}}$, each formed by $r = 3$ NCEs. Each of these NCE cones is represented in a different figure, where the hollow points represent the samples that escape the corresponding cone. The CIP requires that there exists at least one NCE cone that covers all the data. Therefore, the data in this example does not satisfy the CIP.

BASIC (ONEROUS) ALGORITHM

Our construction of NCEs naturally induces an NMF algorithm that can be summarized in the following steps: (i) identifying an orthonormal basis of $\mathcal{W}$, for example, with a singular value decomposition (SVD) of $\mathbf{X}$. (ii) Solving (3) for each $\Omega$ to obtain the collection of all canonical edges $\{\mathcal{E}_{\Omega}\}$. (iii) Checking whether each $\mathcal{E}_{\Omega}$ or $-\mathcal{E}_{\Omega}$ is nonnegative to only keep the set of all NCEs $\{\mathcal{E}_{\Omega}\}_+$. (iv) Finding a subset $\mathbf{W}_{\mathcal{E}}$ of r NCEs among $\{\mathcal{E}_{\Omega}\}_+$ whose cone $\mathcal{C}_{\mathcal{E}}$ contains $\mathbf{X}$, (v) computing $\mathbf{H}_{\mathcal{E}}$ as in (4). The desired NMF is then given by $\mathbf{X} = \mathbf{W}_{\mathcal{E}}\mathbf{H}_{\mathcal{E}}$. By construction, this algorithm is guaranteed to succeed for every $\mathbf{X}$ satisfying the CIP. Obviously, exhaustively computing $\{\mathcal{E}_{\Omega}\}$ and testing r-tuples in $\{\mathcal{E}_{\Omega}\}_+$ brings a heavy computational complexity. There are extreme scenarios where this burden is unavoidable, such as settings where the alignment of $\mathcal{W}$ and the placement of $\mathbf{X}$ over $\mathcal{W}$ is selected adversarially to escape most NCE cones. On the other hand, there are also situations (see orthogonal NMF below) that only require computing a projection operator, bestowing our algorithm a computational complexity comparable to computing an SVD. The computational complexity for most cases lies in between these two extremes, depending on both the orientation of $\mathcal{W}$ and the distribution of $\mathbf{X}$ over $\mathcal{W}$. This basic algorithm is not intended for practical use (except in very small cases). We merely present it for completeness, to build intuition, and because we will use this exhaustive algorithm in Section 4 to learn more about NCEs.

## 3   RELATED WORK

The relevance of NMF cannot be overstated, and is reflected in the plethora of approaches that have studied the problem from a myriad of angles. As consequence, many comprehensive surveys have surfaced over the last decades (Li & ZHANG, 2008; Wang & Zhang, 2012; Huang et al., 2012; Li & Ding, 2018; Gan et al., 2021; Chen et al., 2022; Fathi Hafshejani & Moaberfard, 2022). We refer the reader to these exceptional works and the references therein for a thorough discussion, analysis, and comparison of these approaches. We will, however, discuss one particular approach that stands out because of the similar geometric nature it shares with our work. That is the seminal work of Klingenberg et al. (Klingenberg et al., 2009), whose main idea is to find up to m *extreme vectors* in $\mathbf{X}$ whose cone captures the whole data (see Figure 3). If there exist such vectors, $\mathbf{X}$ is said to satisfy the *extreme data property (EDP)*. The downside of this *extreme vector algorithm* (EVA), besides the high computational cost of finding these extreme vectors, is that the EDP is quite restrictive, because data will rarely satisfy the EDP if $r > 2$ (see Figure 3). Moreover, using m factors instead of r may be overly sufficient, and may even defeat the purpose of NMF, where generally $m \gg r$. This type of overfitting is accentuated when n is comparable to m, resulting in the trivial solution ($\mathbf{W} = \mathbf{X}, \mathbf{H} = \mathbf{I}$) whenever $n \leq m$. In contrast, since NCEs are at the boundary of the nonnegative orthant, NCE cones will generally cover a larger portion of $\mathcal{W}_+$ (see Figure 3-left). In other words, NCE cones are more likely to capture more datasets and generalize better, using fewer factors (see Figure 3-right, and compare to Figure 2). It is worth noting, however, that CIP does not imply EDP, nor viceversa. Finally, an additional advantage of our approach is the much smaller search space. For subspaces in general position, there are at most $\binom{m}{r-1}$ canonical edges, most of which are not NCEs (see Figure 4). In contrast, the EVA requires searching over $\binom{n}{m}$ combinations of samples, and generally $n \gg r$.

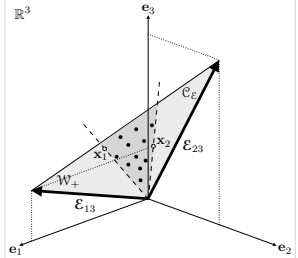 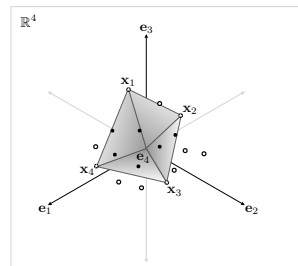

**Figure 3:** The *extreme vector algorithm* (EVA) (Klingenberg et al., 2009) is similar in principle to our approach. EVA aims to find up to $m$ vectors in $\mathbf{X}$ whose cone captures the whole data. **Left:** If $r = 2$, extreme vectors are guaranteed to exist, and EVA is guaranteed to succeed. However, our NCE cone $\mathcal{C}_\mathcal{E}$ covers a larger portion of $\mathcal{W}_+$, and may generalize better. In this example, EVA would not generalize well to new data lying on the lighter shaded region. **Right:** Abstraction of $\mathbb{R}^4$ viewed orthogonally from the $4^{\text{th}}$ dimension. If $r \geq 3$, there may not exist $m = 4$ extreme vectors that capture the rest of the data. Since NCEs are at the boundary of the nonnegative orthant, they are more likely to cover more samples using fewer factors. Compare the fraction of hollow points, representing samples that escape the cone formed by $m = 4$ extreme vectors, with those in Figure 2, which only use $r = 3$ NCEs.

## 4    ON THE AMOUNT OF NCEs AND THE SIZE OF THEIR CONES

Naturally, there exist matrices $\mathbf{X}$ that will not satisfy the CIP. In fact, matrices violating the CIP can be easily constructed adversarially by adding one sample outside each NCE cone (and inside $\mathcal{W}_+$). However, since NCEs are at the boundary where $\mathcal{W}$ crosses the nonnegative orthant, the largest NCE cone will generally cover a large portion of $\mathcal{W}_+$ — and in some cases, all of it (see Section 7). To verify this, we ran an experiment to estimate the amount of NCEs, and the volume of $\mathcal{W}_+$ covered by the largest NCE cone.

In these experiments, we first generated random subspaces according to the uniform distribution over the Grassmannian. This can easily be done by generating bases with standard normal entries. We kept only the subspaces $\mathcal{W}$ that crossed the nonnegative orthant. For each of these subspaces, we generated data with a standard normal distribution over $\mathcal{W}$. Again, this can be easily done creating $\mathbf{UV}$, where $\mathbf{U} \in \mathbb{R}^{m \times r}$ is an orthonormal basis of $\mathcal{W}$, and $\mathbf{V}$ is a matrix with standard normal entries. We only kept the nonnegative columns of $\mathbf{UV}$ to form $\mathbf{X}$. This way the columns of $\mathbf{X}$ have an isotropic normal distribution over $\mathcal{W}_+$. Intuitively, $\mathbf{X}$ generated this way can be thought of as completely random nonnegative data in $\mathcal{W}$. In these experiments, each $\mathbf{X}$ had $n = 10^5$ columns.

**Number of NCEs.** Recall that the number of NCEs, $N$, depends on the orientation of $\mathcal{W}$. To quantify the difficulty of finding the largest NCE cone we recorded $N$ in each trial, which determines the number of NCE cones $K = \binom{N}{r}$. The results for $r \in \{3, 4\}$ are in Figure 4, showing that in general, out of the $\binom{m}{r-1}$ canonical edges that every subspace in general position has, only a handful are nonnegative. This is quite convenient because it means that the search space of the best NCE cones is generally small.

**Size of the Largest NCE Cone.** To quantify the size of the largest NCE cone, we used four notions: (i) the minimum angle between its spanning vectors, quantifying the cone's width, (ii) the volume of the convex hull of the columns of $\mathbf{W}_\mathcal{E}$ and the origin, given by $\sqrt{\det(\mathbf{W}_\mathcal{E}^\mathsf{T} \mathbf{W}_\mathcal{E})}/r!$, quantifying the cone's volume (Ang & Gillis, 2019), (iii) the fraction of nonnegative entries in $\mathbf{H}_\mathcal{E}$, quantifying how many coefficients satisfy the nonnegative constraint (we do not measure $\mathbf{W}_\mathcal{E}$ because recall that $\mathbf{W}_\mathcal{E} \geq \mathbf{0}$ by construction), and (iv) the fraction of nonnegative columns in $\mathbf{H}_\mathcal{E}$, representing the portion of data in $\mathcal{W}_+$ that was captured by the cone, or equivalently, the size of the cone relative to the size of $\mathcal{W}_+$. The larger these measures are, the better, because when cones are larger they can represent more nonnegative data.

We cannot compare against EVA (Klingenberg et al., 2009) because searching over $\binom{n}{m}$ becomes untractable with this large number of samples, and because it would be an uneven comparison, as EVA allows up to $m$ factors, while we only allow $r$ (see discussion in Section 3 for details). Hence, for a vis-à-vis comparison, we measured the same metrics described above (minimum angle, volume, and fraction of nonnegative entries and columns) for the best random cone after $K$ trials, each generated by a subset of $r$ columns in $\mathbf{X}$ (this is similar to what EVA would do, and is essentially what typical algorithms would use as initialization (Fathi Hafshejani & Moaberfard, 2022)). The

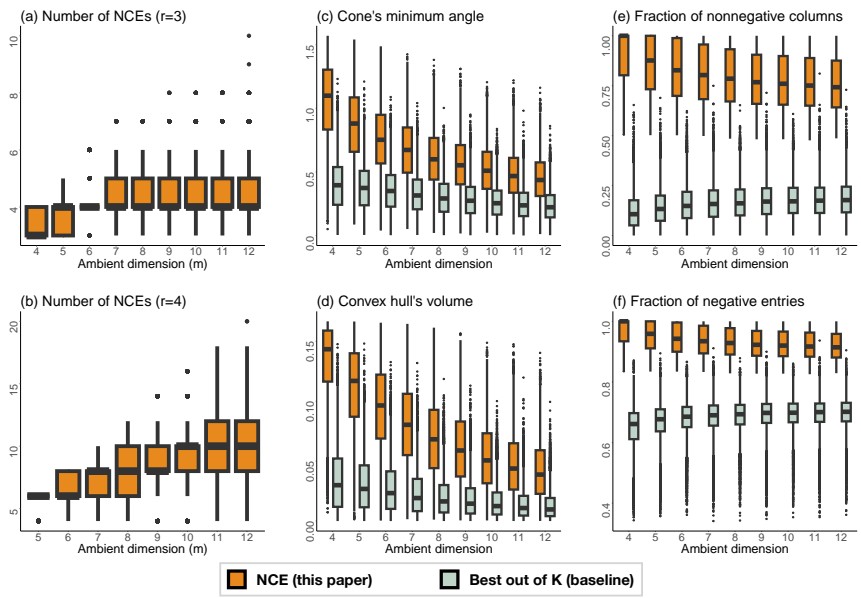

**Figure 4: (a)-(b)** Number of NCEs of subspaces in general position. Out of the $\binom{m}{r-1}$ canonical edges that every subspace in general position has, only a handful of them are nonnegative. This is quite convenient because it means that the search space of NCE cones is generally small. **(c)-(f)** Metrics estimating the size of the largest NCE cone relative to the size of $\mathcal{W}_+$. The larger the better. For baseline comparison we use the best cone out of $K = \binom{N}{r}$, each generated by a random subset of r columns in $\mathbf{X}$. The NCE cone is better by every metric.

results of $10^4$ independent trials with $r = 3$ are summarized in Figure 4, where NCEs are better by every metric.

These experiments show that the largest NCE cone typically covers around 75% of $\mathcal{W}_+$. This is remarkable, especially considering that the nonnegative rank of this data may be higher than r (we cannot know, because learning the nonnegative rank in general is a daunting long open question (Dong et al., 2008)). This means that there may not even exist a rank-r NMF for this data, yet a single NCE cone captures most of it. This suggests that using a few additional NCEs would allow factorizing data whose nonnegative rank is higher than its rank. To test this, we we measured the fraction of columns in $\mathbf{X}$ that were contained in *some* NCE cone. This allowed us to quantify the coverage of $\mathcal{W}_+$ obtained by the NCE cones. We do not provide a box plot for this, because *all* the $10^9$ (1 billion) samples in our experiments were contained in at least one of its corresponding NCE cones. This supports our conjecture that the union of all NCE cones (or equivalently, the cone spanned by all NCEs) is equal to $\mathcal{W}_+$. Such a result would provide a characterization of $\mathbf{W}_+$, and shed some light on learning the positive rank. In practical terms, this would mean that the matrix formed with all the (more than r, yet few) NCEs should allow the desired NMF of any $\mathbf{X} \in \mathcal{W}_+$.

## 5 INEXACT NMF

The results in Figure 4 confirm that NCE cones of subspaces $\mathcal{W}$ in general position cover large portions of $\mathcal{W}_+$, and hence they are likely adequate to represent nonnegative data. Equivalently, many generic datasets $\mathbf{X} \in \mathcal{W}_+$ are likely to satisfy the CIP. However, in general, data may be contaminated with noise or outliers, resulting in observations that may not lie exactly in a subspace. In other words, the observed data may not admit an exact NMF. In that case, the goal of NMF is to identify the nonnegative factors that best approximate the observation given by $\mathbf{Y} = \mathbf{X} + \mathbf{Z}$, where $\mathbf{Z} \in \mathbb{R}^{m \times n}$ represents a noise and outliers matrix. In this case, NMF is typically pursued through a constrained optimization of the form:

$$\min_{\mathbf{W} \in \mathbb{R}^{m \times r}, \mathbf{H} \in \mathbb{R}^{r \times n}} \ell(\mathbf{Y}, \mathbf{WH}) \quad \text{s.t.} \quad \mathbf{W} \geq \mathbf{0}, \ \mathbf{H} \geq \mathbf{0}. \tag{5}$$

Here, $\ell$ can be any appropriate loss. Common choices include the Frobenius norm, the $\ell_1$ norm, the generalized Kullback-Leibler divergence, convex hull volumes, and combinations of these, such as the elastic-net (Li & ZHANG, 2008; Wang & Zhang, 2012; Huang et al., 2012; Li & Ding, 2018;

Ang & Gillis, 2019; Gan et al., 2021; Fathi Hafshejani & Moaberfard, 2022). One natural way to extend our algorithm to this inexact setting is by simply computing the *unconstrained* minimizer

$$(\hat{\mathbf{W}}, \hat{\mathbf{H}}) := \underset{\mathbf{W} \in \mathbb{R}^{m \times r}, \mathbf{H} \in \mathbb{R}^{r \times n}}{\arg \min} \ell(\mathbf{Y}, \mathbf{WH}), \tag{6}$$

and proceeding as before, using $\hat{\mathcal{W}} := \mathrm{span}(\hat{\mathbf{W}})$ in place of $\mathcal{W}$. This approach is guaranteed to work so long as $\hat{\mathbf{X}}$, defined as the projection of $\mathbf{Y}$ onto $\hat{\mathcal{W}}$, satisfies the CIP. In such case, the solution pair $(\hat{\mathbf{W}}_{\mathcal{E}}, \hat{\mathbf{H}}_{\mathcal{E}})$ obtained in this fashion will satisfy the nonnegative constraint, and be optimal in the sense that

$$\ell(\mathbf{Y}, \hat{\mathbf{W}}_{\mathcal{E}} \hat{\mathbf{H}}_{\mathcal{E}}) = \ell(\mathbf{Y}, \hat{\mathbf{W}} \hat{\mathbf{H}}) := \min_{\mathbf{W}, \mathbf{H}} \ell(\mathbf{Y}, \mathbf{WH}) \leq \min_{\mathbf{W} \geq \mathbf{0}, \mathbf{H} \geq \mathbf{0}} \ell(\mathbf{Y}, \mathbf{WH}),$$

where the first equality follows because $\hat{\mathbf{W}}_{\mathcal{E}}$ and $\hat{\mathbf{W}}$ span the same subspace, and the last inequality follows trivially because the minimization is over a smaller (constrained) set. For example, in the particular case of the Frobenius loss, this means that our NMF will be as accurate as PCA.

## 6 NCE CONES AS INITIALIZATION

Per our discussion above, we can conclude that if $\hat{\mathbf{X}}$ satisfies the CIP (and the unconstrained solution to (6) can actually be computed, which is typically the case), then our solution will satisfy the nonnegative constraint, and attain the same loss as the unconstrained solution. However, it is possible that $\hat{\mathbf{X}}$ does not satisfy the CIP. This would be the case if, for example, the data is constructed adversarially, or if the NCE cones of $\hat{\mathcal{W}}$ are too narrow, or if the noise spreads the data too widely over $\hat{\mathcal{W}}_+$. Fortunately, even if $\hat{\mathbf{X}}$ does not satisfy the CIP, NCEs can still be used as starting points to now-classical NMF methods, such as those summarized in (Li & ZHANG, 2008; Wang & Zhang, 2012; Huang et al., 2012; Li & Ding, 2018; Ang & Gillis, 2019; Gan et al., 2021). In fact, one of the most notorious shortcomings of most algorithms is their local minima, and the lack of a definitive initialization strategy (Fathi Hafshejani & Moaberfard, 2022). Precisely because most of the volume of $\mathcal{W}_+$ is covered by NCE cones, we believe NCEs would be good initializations to these methods, which would essentially be doing a refinement to tilt the NCEs in order to capture the few data that escaped their cones.

**Refining the NCE cones.** To test our initialization and refinement strategy described above, we ran an experiment where we generated $\mathbf{X}$ as in Section 4 with $r = 3$ and $m = 10$, then created a noise matrix $\mathbf{Z} \in \mathbb{R}^{m \times n}$ with normal random variables of variance $\sigma^2$, and constructed $\mathbf{Y} = \max(\mathbf{0}, \mathbf{X} + \mathbf{Z})$. Next we identified the largest NCE cone, spanned by $\mathbf{W}_{\mathcal{E}}$, and solved the classic and prevalent alternating Frobenius minimiziation (Paatero, 1997; Lee & Seung, 2000; Li & ZHANG, 2008; Wang & Zhang, 2012; Huang et al., 2012; Li & Ding, 2018; Gan et al., 2021; Fathi Hafshejani & Moaberfard, 2022) until convergence:

$$\mathbf{H}_t = \underset{\mathbf{H} \geq \mathbf{0}}{\arg \min} \|\mathbf{Y} - \mathbf{W}_t \mathbf{H}\|_F \quad \text{and} \quad \mathbf{W}_{t+1} = \underset{\mathbf{W} \geq \mathbf{0}}{\arg \min} \|\mathbf{Y} - \mathbf{WH}_t\|_F, \tag{7}$$

starting with $\mathbf{W}_0 = \mathbf{W}_{\mathcal{E}}$. Figure 5 summarizes the results, where we measure the approximation error on $\mathbf{X}$ and $\mathbf{Y}$, namely

$$\|\mathbf{X} - \mathbf{WH}\|_{\mathbf{F}} / \|\mathbf{X}\|_F, \quad \text{and} \quad \|\mathbf{Y} - \mathbf{WH}\|_{\mathbf{F}} / \|\mathbf{Y}\|_F, \tag{8}$$

for each of the following decompositions: (a) truncated NCE, where the second factor is equal to $\max(\mathbf{0}, \mathbf{H}_{\mathcal{E}})$; this enforces the nonnegative constraint our NCE solution, which is necessary because the data may not satisfy the CIP (there is no need to truncate $\mathbf{W}_{\mathcal{E}}$ because it contains the NCEs, which are nonnegative by construction), (b) NCE + Alternating, where we alternate between the steps in (7), starting with $\mathbf{W}_0 = \mathbf{W}_{\mathcal{E}}$, (c) Alternating, where we alternate between the steps in (7) starting with $r$ random columns from $\mathbf{X}$; we use this classical and prevalent method (Paatero, 1997; Lee & Seung, 2000; Li & ZHANG, 2008; Wang & Zhang, 2012; Huang et al., 2012; Li & Ding, 2018; Gan et al., 2021; Fathi Hafshejani & Moaberfard, 2022) as baseline to test the improvement obtained by our NCE initialization, and (d) PCA, which we use as an idealized (perhaps unattainable) lower bound. The results show that NCEs are as accurate as the Alternating solution, and that when used as initialization, NCEs produce an improved minimum.

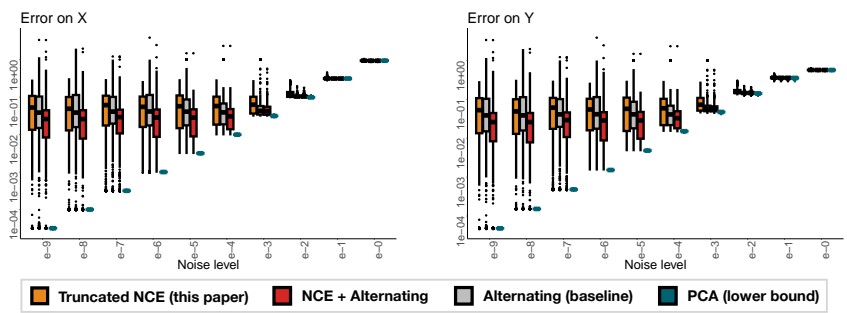

**Figure 5:** Approximation error on the noisy observation $\mathbf{Y}$ and the *true* $\mathbf{X}$. We see that NCEs are directly as accurate as the Alternating solution, and that when used as initialization, they produce an improved minimum.

## 7   THE ORTHOGONAL CASE

Orthogonal NMF (ONMF) is a noteworthy variant of NMF that aims at the same goal of decomposing $\mathbf{X}$ into nonnegative factors $\mathbf{W}$ and $\mathbf{H}$, except that it additionally assumes that $\mathbf{W}^\star$ is orthogonal, and it conversely requires that so is the solution $\mathbf{W}$ (Li et al., 2001; Yuan & Oja, 2005; Ding et al., 2006; Asteris et al., 2015). Motivated by its well-posedness, clustering capabilities, increased interpretability, and success in applications, a plethora of ONMF algorithms have emerged (Choi, 2008; Yang & Oja, 2010; Li et al., 2007a; Cao et al., 2007; Li et al., 2007b; Chen et al., 2009; Pompili et al., 2014; Del Buono & Pio, 2015; vCopar et al., 2019; He et al., 2020; Li & ZHANG, 2008; Wang & Zhang, 2012; Huang et al., 2012; Li & Ding, 2018; Gan et al., 2021; Chen et al., 2022). Most of them modify NMF approaches like (5) to integrate the orthogonality constraint, and can only guarantee local convergence (Asteris et al., 2015; Fathi Hafshejani & Moaberfard, 2022). Here we give a closed-form solution. The key insight is that the orthogonality constraint implicitly requires that the columns of *any* solution $\mathbf{W}$ (including $\mathbf{W}^\star$) have disjoint supports. That is, for any given row, only one column of $\mathbf{W}$ may be nonzero. This means that each column of $\mathbf{W}$ must lie in *some* canonical face $\mathcal{F}_\Omega$, and must therefore be an NCE. The challenge, as in the non-orthogonal case, lies in identifying the sets $\Omega$ corresponding to the NCEs in $\mathbf{W}$. Fortunately, in the orthogonal case there is a trivial way to identify these sets $\Omega$ through the projection operator onto $\mathcal{W}$:

$$\mathbf{P}_\mathcal{W} \;=\; \mathbf{X}(\mathbf{X}^\mathsf{T}\mathbf{X})^{-1}\mathbf{X}^\mathsf{T}.$$

Since projection operators are unique, $\mathbf{P}_\mathcal{W}$ must be equal to $\mathbf{W}\mathbf{W}^\mathsf{T}$, where $\mathbf{W}$ is an orthonormal basis of $\mathcal{W}$. Since $\mathbf{W}$ has r columns with disjoint supports, the columns (and rows) of $\mathbf{P}_\mathcal{W}$ can only have r distinct supports, which coincide with the supports of the columns of $\mathbf{W}$. Moreover, since $\mathbf{P}_\mathcal{W}$ is rank-r, the columns of $\mathbf{P}_\mathcal{W}$ with the same support must be co-linear. Hence, we can construct our first factor $\mathbf{W}_\mathcal{E}$ as the matrix with any r columns of $\mathbf{P}_\mathcal{W}$ with distinct supports, normalized, so that our second factor, given by (4) simplifies to $\mathbf{H}_\mathcal{E} = \mathbf{W}_\mathcal{E}^\mathsf{T}\mathbf{X}$.

**Noisy ONMF.** In the same spirit of Section 5, whenever the data does not admit an exact ONMF, one can generalize the ONMF algorithm above using the following straightforward steps. First, identify the unconstrained solution $\hat{\mathbf{W}}$ to the approximation in (6), for example, using an SVD in the case of the Frobenius norm loss. Then compute the projection operator onto $\hat{\mathcal{W}}$, given by

$$\mathbf{P}_{\hat{\mathcal{W}}} \;=\; \hat{\mathbf{W}}(\hat{\mathbf{W}}^\mathsf{T}\hat{\mathbf{W}})^{-1}\hat{\mathbf{W}}^\mathsf{T}.$$

Next, estimate the r disjoint supports in $\mathbf{P}_{\hat{\mathcal{W}}}$, either by inspection using a threshold parameter related to the noise level, or through another method like spectral clustering. For each estimated support $\Omega$, identify the vectors $\hat{\mathbf{w}}^\Omega \in \mathbb{R}^{|\Omega|}$ and $\hat{\mathbf{h}} \in \mathbb{R}^n$ whose outer product minimizes the loss on the restriction of $\mathbf{Y}$ to the coordinates in $\Omega$, i.e.,

$$(\hat{\mathbf{w}}^\Omega, \hat{\mathbf{h}}) \;=\; \underset{\mathbf{w}^\Omega \in \mathbb{R}^{|\Omega|}, \mathbf{h} \in \mathbb{R}^m}{\arg\min} \; \ell(\mathbf{Y}^\Omega, \mathbf{w}^\Omega\mathbf{h}^\mathsf{T}).$$

For example, in the Frobenius case, $\hat{\mathbf{w}}^\Omega$ would be the leading left singular vector of $\mathbf{Y}^\Omega$. Let $\hat{\boldsymbol{\mathcal{E}}}_\Omega$ be the vector in $\mathbb{R}^m$ that is equal to $\hat{\mathbf{w}}^\Omega$ in the entries of $\Omega$, and zero elsewhere. Notice that $\hat{\boldsymbol{\mathcal{E}}}_\Omega \in \mathcal{F}_\Omega$ by construction. Finally, our approximate ONMF is given by the $m \times r$ matrix $\hat{\mathbf{W}}_\mathcal{E}$ formed with $\{\hat{\boldsymbol{\mathcal{E}}}_\Omega\}$ as columns, and $\hat{\mathbf{H}}_\mathcal{E} = \hat{\mathbf{W}}_\mathcal{E}^\mathsf{T}\mathbf{Y}$.

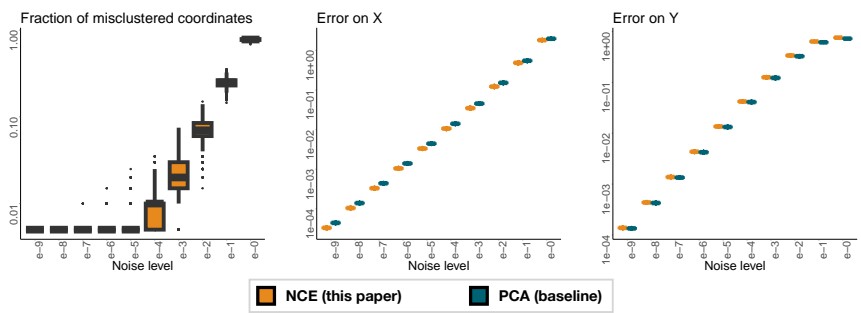

**Figure 6: Left:** Recovery error on the disjoint supports $\Omega$. **Center:** Approximation error on the *true* $\mathbf{X}$. **Right:** Approximation error on the noisy observation $\mathbf{Y}$.

## A SURPRISE

To test our ONMF approach, we ran an experiment where we generated $\mathbf{W}^{\star}$ as the normalized version of a matrix whose columns have random disjoint supports of equal size (up to rounding errors), populated with nonnegative standard normal entries. This way $\mathbf{W}^{\star}$ is orthogonal and nonnegative, according to the ONMF model. Next we populated $\mathbf{H}^{\star} \in \mathbb{R}^{r \times n}$ with nonnegative standard normal entries, and $\mathbf{Z} \in \mathbb{R}^{m \times n}$ with normal entries of variance $\sigma^2$. We constructed $\mathbf{Y} = \max(0, \mathbf{X} + \mathbf{Z})$, and factorized it as described above, using the Frobenius loss $\|\mathbf{Y} - \mathbf{W}\mathbf{H}\|_F$ in (6), and spectral clustering directly on $\mathbf{P}_{\hat{\mathcal{W}}}$ to estimate the disjoint supports. The results of $10^4$ trials with $m = n = 200$ and $r = 10$ are summarized in Figure 6, where we measure the fraction of misclustered coordinates in the support sets $\Omega$, and the errors on $\mathbf{X}$ and $\mathbf{Y}$, as in (8). For comparison, we use PCA as baseline, whose reconstruction accuracy (being unconstrained) surpasses that of any known ONMF algorithm (Choi, 2008; Yang & Oja, 2010; Li et al., 2007a; Cao et al., 2007; Li et al., 2007b; Chen et al., 2009; Pompili et al., 2014; He et al., 2020; Li & ZHANG, 2008; Wang & Zhang, 2012; Huang et al., 2012; Del Buono & Pio, 2015; vCopar et al., 2019; Li & Ding, 2018; Ang & Gillis, 2019; Gan et al., 2021; Lee & Roy, 2021; Baur et al., 2022; Chen et al., 2022; Asteris et al., 2015; Fathi Hafshejani & Moaberfard, 2022). In other words, if we do better than PCA, we do better than any state-of-the-art ONMF algorithm, and it is not necessary to compare against any other.

Interestingly, these results show that our ONMF approach yields a factorization $(\hat{\mathbf{W}}_{\mathcal{E}}, \hat{\mathbf{H}}_{\mathcal{E}})$ that, besides being nonnegative, is slightly more accurate than the unconstrained solution $(\hat{\mathbf{W}}, \hat{\mathbf{H}})$ to (6) in the sense that

$$\ell(\mathbf{X}, \hat{\mathbf{W}}_{\mathcal{E}}\hat{\mathbf{H}}_{\mathcal{E}}) \ \leq \ \ell(\mathbf{X}, \hat{\mathbf{W}}\hat{\mathbf{H}}).$$

This should be surprising for two reasons: (a) unconstrained solutions are generally as good as constrained solutions, and often better, as they are allowed to search over a larger space, and (b) the unconstrained solution is supposed to be *the* optimal low-rank factorization. The reason behind these surprising results is that we are using the orthogonality and nonnegativity constraints as side information to improve our estimates. Thanks to these constraints, we know that the restriction of $\mathbf{W}^{\star}$ to the coordinates in the support set $\Omega$ of any of its columns spans only a line (because such restriction of $\mathbf{W}^{\star}$ is zero in all its remaining columns). We exploit this knowledge when we construct $\hat{\mathbf{\mathcal{E}}}_{\Omega}$, similar to the refinement step of *lasso* after identifying the sparse support. If the support $\Omega$ is estimated correctly, then $\hat{\mathbf{\mathcal{E}}}_{\Omega}$ is by construction the line that best approximates $\mathcal{W}$ in the coordinates of $\Omega$, and ignores the remaining coordinates, filling them with zeros. This induces zero error on the remaining coordinates, where the corresponding column of $\mathbf{W}^{\star}$ is known to be zero. In contrast, the unconstrained estimates are affected by (and account for) the noise in all coordinates. That is why

$$\ell(\mathbf{Y}, \hat{\mathbf{W}}\hat{\mathbf{H}}) \ \leq \ \ell(\mathbf{Y}, \hat{\mathbf{W}}_{\mathcal{E}}\hat{\mathbf{H}}_{\mathcal{E}}),$$

which can be seen directly in Figure 6. In words, this means that the unconstrained estimate is approximating both the nonnegative data of interest $\mathbf{X} = \mathbf{W}^{\star}\mathbf{H}^{\star}$, and the noise $\mathbf{Z}$. Consistent with fact (2), this results in a better overall approximation of $\mathbf{Y} = \mathbf{X} + \mathbf{Z}$ than our ONMF. However, since we are only interested in approximating $\mathbf{X}$, and not the noise, we conclude that our ONMF estimator is better. To summarize, we are only able to improve on the unconstrained estimator because we know the form of $\mathbf{W}^{\star}$, and we use this side information to our advantage. Hence, our results do not contradict facts (a) and (b).

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
