# OpenReview forum: "Nonnegative Matrix Factorization through Canonical Edges"
_ICLR.cc/2024/Conference — Submitted to ICLR 2024_

### Official Review · Reviewer_qZPt · 2023-10-17

**Soundness:** 2 fair
**Presentation:** 1 poor
**Contribution:** 1 poor
**Rating:** 3
**Confidence:** 4

**Summary:**

In this work, the authors claim to propose a novel approach for NMF by introducing the concept of nonnegative canonical edges (NCEs). The approach gives a closed-form solution to a special case of NMF where the factors are orthogonal and provides a deterministic optimal solution to another special case of NMF where the data is close to the cone formed by the NCEs.  The authors also claimed to provide an improved initialization for classical NMF algorithms in general.

**Strengths:**

The introduced concept of NCEs and the corresponding approach may be of interest.

**Weaknesses:**

1. There are significant exaggerated statements in this submission, and it appears the authors are not well-versed in the literature on NMF.

- In the first paragraph of page 2, the authors claim that "In doing so, we provide, to the best of our knowledge, the first ONMF method with global guarantees, thus answering an over 20-year-old open question." This claim appears to be unaware of the seminal work (Arora et al., 2012) and it follow-up works such as (Asteris et al., 2015) and (Charikar & Hu, 2021), which provide global guarantees for NMF or ONMF methods. Moreover, for such a claim, I would expect to see some recovery bounds (for the inexact case) or formal proofs for the global optimality. However, I cannot find any theoretical guarantee in this submission.

Arora, S., Ge, R., Kannan, R., & Moitra, A. (2012, May). Computing a nonnegative matrix factorization--provably. In Proceedings of the forty-fourth annual ACM symposium on Theory of computing (pp. 145-162).

Asteris, M., Papailiopoulos, D., & Dimakis, A. G. (2015). Orthogonal NMF through subspace exploration. Advances in neural information processing systems, 28.

Charikar, M., & Hu, L. (2021, March). Approximation algorithms for orthogonal non-negative matrix factorization. In International Conference on Artificial Intelligence and Statistics (pp. 2728-2736). PMLR.

- The authors claim in the abstract that "NCEs provide an improved initialization for classical NMF algorithms in general". But their limited and weak experiments on synthetic data are clearly not sufficient to support this claim.

2. This submission is written in a terrible form. For example:

- The first paragraph of **Contributions** is confusing and lacks precise definitions of key concepts. It would be more effective to defer the detailed discussion of these concepts to later sections and summarize the main contributions concisely.

- The authors are unable to provide a mathematical characterization or experimental evidence to substantiate the strength of the canonical inclusion property (CIP), and the concepts introduced in the paper (NCEs and CIP) do not appear practically relevant to me.

**Questions:**

See the Weaknesses.

---

### Official Review · Reviewer_bRYi · 2023-10-20

**Soundness:** 2 fair
**Presentation:** 2 fair
**Contribution:** 2 fair
**Rating:** 5
**Confidence:** 3

**Summary:**

This paper presents a novel approach to nonnegative matrix factorization (NMF) by introducing the concept of nonnegative canonical edges (NCEs). These NCEs are intersections of the principal subspace containing the data to be factored with canonical faces of the nonnegative orthant. Through this lens, the proposed approach yields a closed-form solution to the special NMF case where (at least one of) the factors are required to be orthogonal. In the general NMF case, NCEs provide a deterministic optimal solution whenever the data resides within or in proximity to the cone formed by the NCEs. Furthermore, NCEs provide an improved initialization for classical NMF methods in general.

**Strengths:**

1. The authors give the open questions in the Introduction part, and give simple strategies that use NCEs in the presence of noise.

2. The paper is clearly organized by proposing several fundamental questions in the Contributions part of Introduction, which make the authors better understand the organization of this paper.

**Weaknesses:**

1. The major problem of this paper is the limited novelty in the formulation based on the proposed NCEs and CIP. The motivation of constructing canonical edge is not clearly stated. The canonical inclusion property is not well supported by the theoretical illustration.

2. The authors do no clear stated why the number of NCEs should be studied in Section 4.

3. In Section4, the evidence that NCE cons cover the large portions is not clearly given based on the given number of NCEs and size of the largest NCE Cone.

4. The performed experiments for validation in this paper are not sufficient, i.e., the experiments conducted for studying the number of NCEs of subspaces in general position in Figure 4.

**Questions:**

What are the experimental settings for different sections in this paper? The authors are supposed to clearly list them in the corresponding part of the paper.

---

### Official Review · Reviewer_VKXr · 2023-10-31

**Soundness:** 2 fair
**Presentation:** 1 poor
**Contribution:** 2 fair
**Rating:** 5
**Confidence:** 3

**Summary:**

This paper presents a new variant of nonnegative matrix factorization that factorizes the data under the so-called canonical inclusion property, namely that the obtained subspace intersects the canonical faces of the nonnegative orthant.

**Strengths:**

The paper tackle a novel problem and provides some illustrations to better understand this work.

**Weaknesses:**

It is not clear why one needs to investigate the proposed canonical edges for nonnegative matrix factorization. The paper does not provide any motivation for this work.

Another major issue in this paper is that the proposed method is not compared to any other method from the literature. There is no experimental results on real data. The only experiments, done on some too simple synthetic data, are roughly for sanity check.

It is difficult to understand the conducted experiments, and the supplementary materials do not help (single main file)

What does “Le pièce de résistance” mean ?

There are some spelling and grammatical errors, such as “an optimal solution that it is”, “This suggest that”, “Sections 5 and 6 discusses”, “we we measured the”, “alternating Frobenius minimiziation”

**Questions:**

What is the motivation of this work ? can you provide some applications that have some interest in solving the investigated problem.

Why there is no experiments on real data ?

Why there is no comparative analysis to demonstrate the relevance of the proposed method and to position its performance within the literature ?

---

### Meta-Review · Area_Chair_S8xu · 2023-12-07

**Metareview:**

This paper proposes to solve the NMF problem by seeking the so-called nonnegative canonical edges.

**Justification For Why Not Higher Score:**

The idea is not new. Some assumptions are highly related to the so-called "separable NMF". In addition, the number of NCEs could be exponentially many, which makes it impractical for general NMF.

**Justification For Why Not Lower Score:**

N/A.

---

### Decision · Program_Chairs · 2024-01-16

Reject